# Real-life evaluation of histologic scores for Ulcerative Colitis in remission

**Christian Børde Arkteg** [1]*, **Sveinung Wergeland Sørbye** [2], **Lene Buhl Riis** [3,4], **Stig Manfred Dalen** [2], **Jon Florholmen** [1,5], **Rasmus Goll** [1,5]

1 Research Group Gastroenterology Nutrition, Institute of Clinical Medicine, UiT The Arctic University of Norway, Tromsø, Norway, 2 Department of Clinical Pathology, University Hospital of North Norway, Tromsø, Norway, 3 Department of Pathology, Herlev-Gentofte Hospital, Herlev, Denmark, 4 University of Copenhagen, Copenhagen, Denmark, 5 Department of Gastroenterology, Division of Internal Medicine, University Hospital of North Norway, Tromsø, Norway

* car009@post.uit.no

## Abstract

### Background

Histological evaluation of ulcerative colitis (UC) patients has been debated ever since the first description of the disease and its role in follow-up has never been fully established. Recent evidence suggests an added benefit in accuracy when evaluating if the patient is in remission. Unfortunately, there are several different histological indices, and it is difficult to compare outcomes where different scores are applied. Histopathological evaluation is prone to subjective biases, despite the use of indices. In addition, these indices are developed by expert IBD pathologist, but applied at large, by general pathologist. Therefore, we evaluated the three most applied histological indices for UC on samples from patients in remission to compare test qualities and estimate their usefulness to identify remission by both general and GI specialized pathologist.

### Method

Mucosal biopsies from 41 UC patients in clinical and endoscopic remission were collected as part of a larger study on UC. Three pathologists blinded to the patients' clinical status evaluated them using Geboes score (GS), Nancy Index (NI) and Robarts Histopathological Index (RHI). We calculated the agreement between the pathologists using Inter-class correlation (ICC) and visualized it with ICC-plots and Bland-Altman plots. Association between clinical factors and histological category were analysed by Fisher's exact test.

### Results

The ICC value for GS, RHI and NI were 0.85, 0.73 and 0.70 respectively. The limits of agreement were ±6.1, ±4.0 and ±1.4, for GS, RHI and NI, respectively. Mayo endoscopic subgrade and UC clinical score did not show association with any histological scores. Despite clinical and endoscopic remission 7–35% of the patients displayed histological inflammation on a level classified as active disease, depending on the index and cut-off.

**Data Availability Statement:** All relevant data are within the paper and its Supporting Information files.

**Funding:** This work was funded by Northern Norway Regional Health Authority ID: SFP1134-13,

HNF1517-20 and HNF1468-19 The publication charges for this article have been funded by a grant from the publication fund of UiT The Arctic University of Norway The funders had no role in study design, data collection and analysis, decision to publish, or preparation of the manuscript.

**Competing interests:** The authors have declared that no competing interests exist.

## Conclusion

A substantial amount of UC patients in clinical and endoscopic remission display inflammation on a histological level, but the ability to classify these patients accurately and consistently could be improved.

## Introduction

Ulcerative colitis (UC) is a chronic disease of the colon with relapsing-remitting characteristics. The introduction of targeted antibodies, such as anti-TNF, directed against key pro-inflammatory mediators, has improved patient outcome and lowered colectomy rates [1, 2]. However, the medication is expensive and has serious side effects like lowering the immune competency against certain infections and cancers [3]. Therefore, finding optimal criteria for remission is important, not only for the patients' health but also in a health-care economic aspect.

There is no universally applied definition of the state of remission, but usually only clinical or endoscopy-based scores are applied. The current treatment goal is partial Mayo score/ SCCAI $\leq 1$ and mucosal healing (MH) which is defined by Mayo endoscopic score (MES) of $\leq 1$ [4–6]. However, this recommendation is moving towards MH to include only MES/Ulcerative Colitis Endoscopic Index of Severity (UCEIS) of 0 [7].

Histology adds a dimension in the evaluation of remission which can be beneficial. This was illustrated by a relapse prediction model that included both histologic and endoscopic activity. The model could predict relapse better than endoscopy alone [8]. Histology can detect subclinical inflammation despite endoscopically normal/near-normal mucosa and this inflammation increases the risk of an unfavourable outcome, such as relapse or neoplasia [9–13]. The European Crohn's and Colitis organization has recently published guidance on this topic [14]. However, the multiple scoring indices for histopathology in UC makes it difficult to compare the results between papers [15]. In addition, most of these indices lack thorough validation [16, 17]. Geboes Score (GS), Nancy Index (NI) and Robarts Histopathological Index (RHI) are the few that are partly/fully validated, and they vary in complexities and features they evaluate [18–20]. The position paper for ECCO recommends NI for clinical practice and observational studies. For histology to be of use in determining remission certain criteria must be fulfilled: A. It must add information of the inflammatory state not otherwise obtained. B. It must reliably and accurately identify these signs. C. The use yields a benefit in patient outcome. This paper focuses on the two first subjects, but also explores the relationship between histology grades and clinical parameters.

The US Food and Drug administration now recommends that histopathology should be included as endpoints in new trials. Therefore, there is an urgent need to define the histopathological remission state so it can be applied in trials and in the clinic. To address this, we evaluated the properties of the three most validated histological indices in a population defined to be in remission according to the current recommendations.

## Material and methods

### Study population

This study is a part of the Advanced Study of Inflammatory Bowel disease (ASIB) prospective study at the University Hospital of Northern Norway, Tromsø. All study participants gave

written, informed consent. The study and storage of biological material was approved of by the Regional Committees for Medical and Health Research Ethics, division North (REK Nord ID:2012/1349).

The selected participants were previously diagnosed with UC according to diagnostic recommendation [5]. Overview of baseline characteristics is presented in Table 1. Sample collection was performed at routine endoscopy for patients in remission from August 2013 to April 2016.The most frequent clinical indication being follow-up due to cancer screening and de-escalation of treatment. Inclusion criteria were age between 18 and 80 with clinical and endoscopic remission defined as Mayo clinical score/Ulcerative Colitis Clinical Score (UCCS) of 0 or 1 and Mayo endoscopic score (MES) of 0 or 1. Total Mayo score above 1 or rectal bleeding was not included. IBD medication was not an inclusion or exclusion factor.

## Histology

All biopsies were formalin fixed immediately after sampling and embedded in paraffin. Multiple 3-μm sections were cut with a Micron microtome (HM355S, ThermoFisher, Tudor Rd, Runcorn WA7 1TA, United Kingdom) and stained with haematoxylin and eosin. In cases of multiple biopsies from one patient, the highest scoring biopsy was included in the analysis. Slides were investigated by three pathologists (SWS, SMD and LBR) blinded to the endoscopic score and biopsy location. SWS and SMD are general pathologists who evaluate 200–300 GI samples yearly, of which about 20–30 are IBD related. LBR works mainly with GI samples and sees around 180–360 IBS samples yearly. The final score for a biopsy is the average of the three pathologists. Two of the pathologists are located at the University Hospital of North-Norway while the third is located at Herlev Hospital in Denmark. SWS and LBR evaluated the slides using white light microscopy, while SMD evaluated the slides digitally, scanning them with Pannoramic 250 Flash III (3DHISTECH Ltd. Budapest, Öv u. 3, 1141, Hungary) at 40x with CaseViewer 2.3. In order to evaluate for intra-rater variability and explore the difference between light microscopy and digital microscopy SWS evaluated the slides a second time digitally with a 2-month interval. All pathologists were sent a scoring protocol to improve coherent rating.

The definition of remission across the three indices is not set, different studies have used different cut-offs. GS range from 13 to 7 (Table A of GS continuous vs original Table A in S1 Appendix) and RHI from <6 to ≤1 [21–26]. While the developers of NI suggest that ≤1 should be the cut-off, 0 is also applied in some papers [27]. As the cut-off values are debatable, it is of interest to explore the impact these definitions would have on a population in clinical/endoscopic remission. Therefore, we defined two separate definitions of histological remission,

**Table 1. Baseline characteristics.**

| UC remission | |
|---|---|
| Number of patients | 41 |
| Gender(M/F) | 16/25 |
| Age(mean) | 43 |
| Biopsy location (Rectum/Sigmoid/Other) | 22/13/6 |
| Average endoscopic score (MES) | 0.24 |
| Average clinical score (UCCS) | 0.15 |
| Median Robarts Histopathological Index | 1 |
| Median Nancy Index | 0 |
| Median Geboes Score | 4 |
| Average Disease duration | 8.8 years |

one strict and one relaxed. To be in line with previous research and to exclude mucosal neutrophils and basal plasmacytosis, the strict cut-off was GS <7, RHI <4 and no points allowed for neutrophils in neither epithelium nor lamina propria and NI = 0 [14, 27–30]. The relaxed cutoffs for remission for NI and RHI are the developer's recommendation (NI <2, RHI <6). For GS, the relaxed cut-off is widely applied (GS <13) [30].

## Statistics

All statistics were performed with Rstudio Version 1.2.5019. Inter/intra-rater calculation was done with the "irr" and "KappaGUI" packages. Inter-rater on ordinal/continuous variables was performed with two-way random, average score, intraclass correlation coefficient (ICC) for consistency (C,3). Intra-rater was performed with two-way random, single score ICC for absolute agreement (A,1). On categorical variables Fleiss' kappa was applied and evaluated according to Landis et. al: < 0: Poor agreement, 0.01–0.20: Slight agreement, 0.21–0.40: Fair agreement, 0.41–0.60: Moderate agreement, 0.61–0.80: Substantial agreement, 0.81–1.00: Almost perfect agreement [31]. Bland Altman plots were calculated with mean squared error according to method proposed by Mark Jones et.al [32]. All scores were standardized by dividing them on their theoretical max and then transformed with square root because of skewness in the raw data. The standardization makes limits of agreement (LOA) directly comparable. We investigated systematic rating differences between raters with Kruskal-Wallis rank sum test. If significant, we made a sub-analysis to identify which graders were different. The subanalysis was performed as pairwise comparisons using Wilcoxon rank sum test with multiple comparison adjusted p-values (Benjamini and Hochberg). Relationships between two dichotomous variables was assessed with chi-square test or Fisher exact test, dependent on group sizes. These statistical tests were performed with "rstatix" package for R.

## Results

In total 41 biopsies from 41 UC patients in clinical and endoscopic remission were evaluated by two general pathologists and one GI-specialized pathologists using all three scoring indices. Only five biopsies were evaluated as 0 by all three pathologists across all three indices. Median scores for indices were 7, 4 and 1 for GS, RHI and NI, respectively. Between 7 and 15% of all the samples still exhibited histological activity to such a degree that they would be classified as active disease with a relaxed histological remission definition (GS<13, R<5, N<2). With a stricter remission definition, the share of active disease increases to between 22–32% of all samples (GS<7, R<4, N<1). There was a systematic difference between the three pathologists, where LBR rated higher on average than SWS and SMD with GS, but with a similar standard deviation (S1 Table). This was significant in a Kruskal-Wallis rank sum test for both GS and RHI (S1 Fig and S2 Table)

### Agreement between raters

The inter-rater ICC value for the features vary from poor(<0.50) to excellent (>0.90) according to the classification suggested by Koo et al. (Table 2) [33]. Features describing severe inflammations are over/under-estimated due to small sample size for those features. Only GS achieves an agreement of good, while RHI and NI achieves moderate agreement. The intrarater evaluation displayed better results, as the final score for the three indices ranged from good to excellent (0.78–0.92, Table 2). Fig 1 is an ICC plot illustrating inter-rater agreement between raters for each slide on the Final score for each index. Modified Bland-Altman (BA) plots displayed the limit of agreement as ±0.53, ±0.59 and ±0.35 for GS, NI and RHI, respectively (Fig 2). If transformed back to the original values it corresponds to ±6.1, ±1.4 and ±4.0.

**Table 2. ICC values for histological feature agreement.**

| Geboes Score | ICC Inter | ICC Intra | N. patient* |
|---|---|---|---|
| Grade 0 Structural architectural changes | 0.65 (0.42–0.80) | 0.95 (0.91–0.97) | 26 |
| Grade 1 Chronic inflammatory infiltrate | 0.83(0.71–0.90) | 0.78(0.62–0.88) | 26 |
| Grade 2A Eosinophils | 0.65 (0.42–0.80) | -0.04(-0.33–0.26) | 15 |
| Grade 2B Lamina propria neutrophils | 0.77(0.61–0.87) | 0.89(0.80–0.94) | 4 |
| Grade 3 Neutrophils in epithelium | 0.89(0.81–0.94) | 1.00 | 4 |
| Grade 4 Cryptdestruction | -0.03(-0.73–0.42) | 0.00 (-0.30–0.30) | 3 |
| Grade 5 Erosion/ulcus | 0.10 (0.51–0,49) | NA- | 3 |
| Final Grade | 0.85(0.75–0,91) | 0.96 (0.93–0.98) | 35 |
| Robarts Histopathological Index | | | |
| Chronic Inflammatory Infiltrate | 0.83 (0.71–0.90) | 0.77(0.62–0.87) | 26 |
| Lamina propria neutrophils | 0.77(0.61–0.87) | 0.79(0.64–0.88) | 4 |
| Neutrophils in epithelium | 0.89(0.81–0.94) | 1.00 | 4 |
| Erosion/Ulceration | 0.09(0.53–0.48) | NA- | 3 |
| Final Grade | 0.73(0.54–0.85) | 0.96(0.93–0.98) | 26 |
| Nancy Index | | | |
| Chronic inflammatory cell | 0.42(0.02–0.67) | 0.38 (0.10–0.61) | 5 |
| Acute inflammatory cells | 0.79 (0.64–0.88) | 0.86 (0.75–0.92) | 10 |
| Ulceration | -0.04(-0.74–0.41) | NA | 2 |
| Final Grade | 0.70(0.50–0.83) | 0.86(0.75–0.92) | 13 |

*Number of patients with a score >0.

There is a tendency of higher agreement in the extremes of the scores, albeit not a big difference.

## Remission aid and clinical application

Next, we evaluated the inter-rater properties with two different cut-offs for remission, relaxed (GS <13, RHI <5, NI <2) and strict (GS <7, RHI<3 and NI <1). The latter definition resulted in a doubling of patients defined with active disease with GS and NI but no change in RHI (S2 Fig). Both cut-offs showed similar kappa values, from fair to moderate agreement (Table 3). NI and RHI performed slightly better than GS with strict cut-off.

Thereafter, we investigated if there was a difference between high (MES = 1 and UCCS = 1) and low (MES = 0 and UCCS = 0) endoscopic and clinical grade and the histological category (Active or Remission). Neither clinical grade, nor endoscopic grade showed significant dependence with the histological category, regardless of strict or relaxed cut-off (S3 Table).

To control for potential confounding factors, we investigated difference in histology score by their biopsy location and IBD medication. The distribution was rectum (n = 22), sigmoid (n = 13), and other (n = 6). The Kruskal-Wallis test showed no significant difference between the different locations (S3 Fig) and the Wilcoxon rank sum test showed no effect of different medication on the histological scores (S4 Table).

## Discussion

This observation study evaluates the performance of the three most validated histological scores for UC in a remission setting. The main findings are a poor to excellent inter-rater agreement between the three histological scores, as well as a fair to moderate inter-rater agreement for determining remission. The patients were defined as remission patients according to

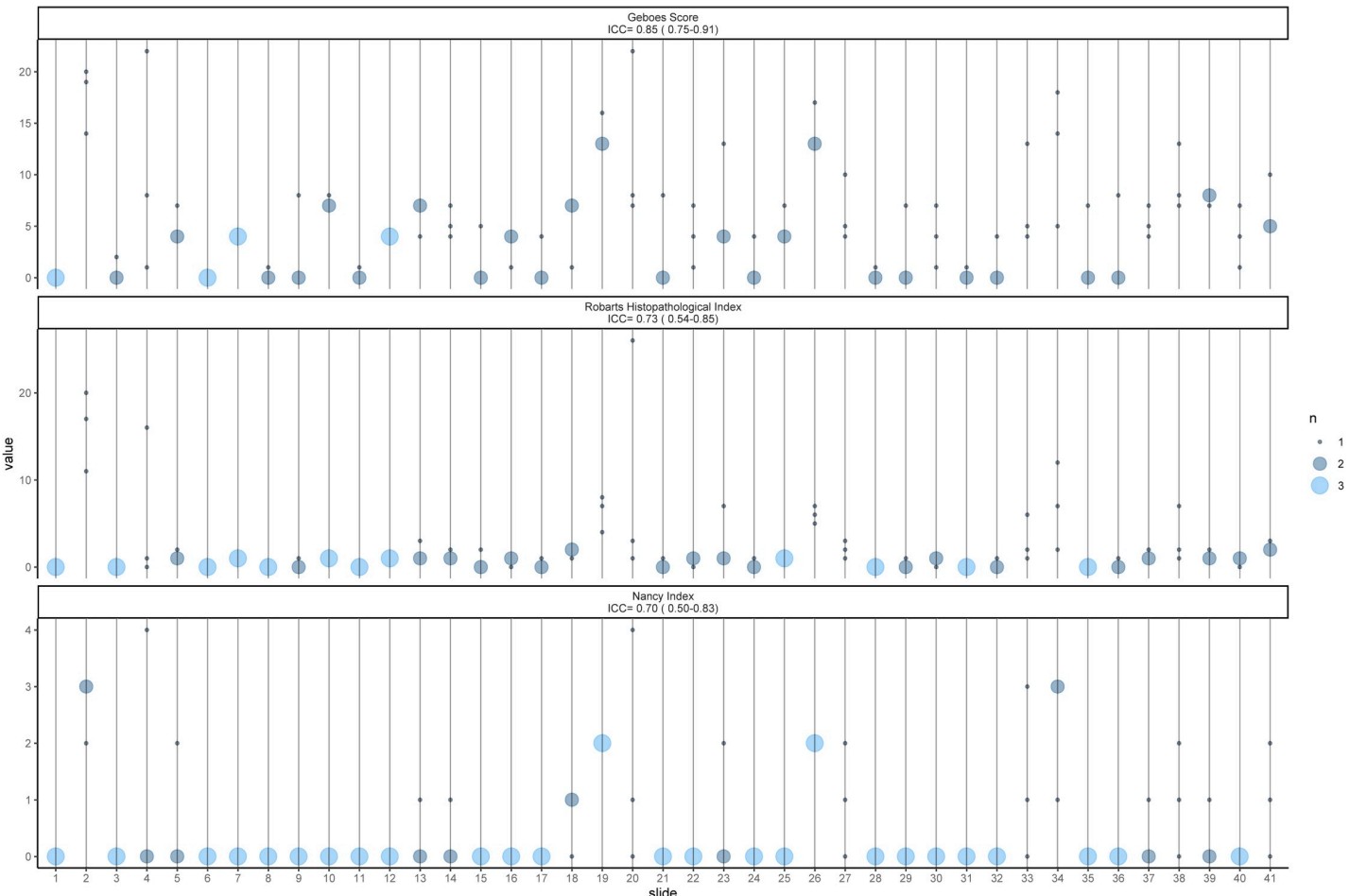

**Fig 1. ICC dot plot.** The plot illustrates how the Final grade for each slide is scored by the three pathologists. The size of circle indicates how many raters gave the same score.

the current guidelines (i.e. clinical and endoscopic remission). Nevertheless, a substantial number showed histologic inflammatory activity, indicating that histology can unveil inflammatory features in a population of patients in remission pre-selected on clinical and endoscopic findings. This is in line with previous publications [34, 35].

Compared to previous research, our results show lower concordance between raters. Jairath et al. had inter-rater ICC of 0.88, 0,86, 0,80 for GS, RHI and NI respectively [36] and Marchal-Bressnot et al. achieved a ICC value of 0.86 when developing the NI [20]. Mosli et al. achieved 0.82 when developing RHI [19]. The GS method paper applied pairwise Cohen's kappa and is not comparable with our results [18]. This difference could be either the result of different interpretations of scores between our raters or observational errors. All raters were sent the same scoring protocol (S1 Appendix) to improve coherent rating. It could be argued that a scoring protocol is not sufficient to ensure coherent rating from general pathologists. We argue that this the actual situation in most hospitals outside specialized tertiary centres. Thus, our results show the real-life utility of the scores. By including nothing but patients in remission, only the lower range of the histologic scales are represented, and this may be viewed as a "stress test" of the scores for this specific patient group. Consequently, lower inter-rater agreement is expected.

# Bland-Altman plot

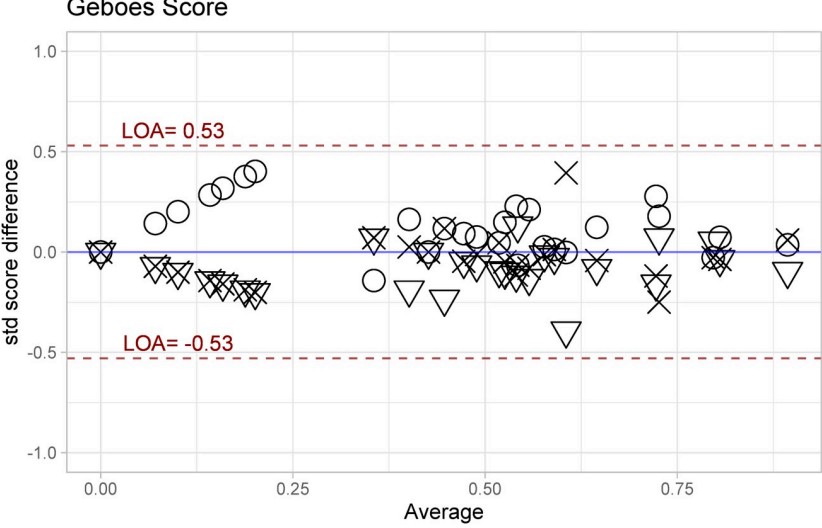

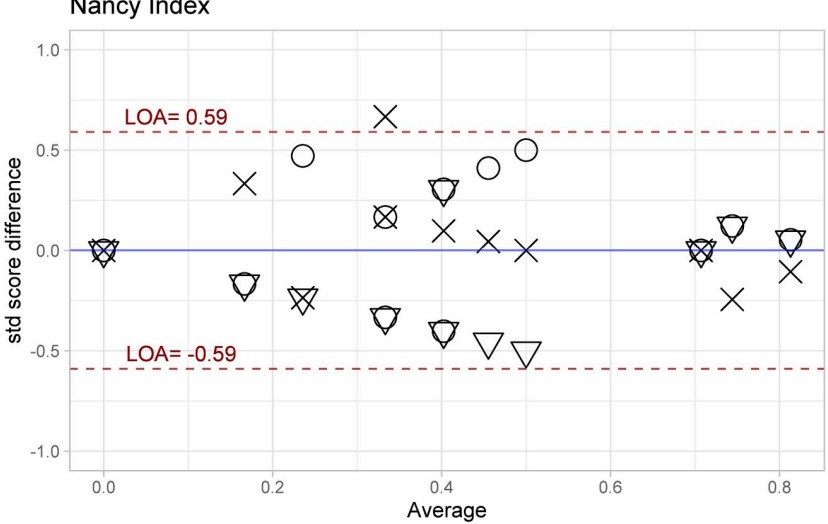

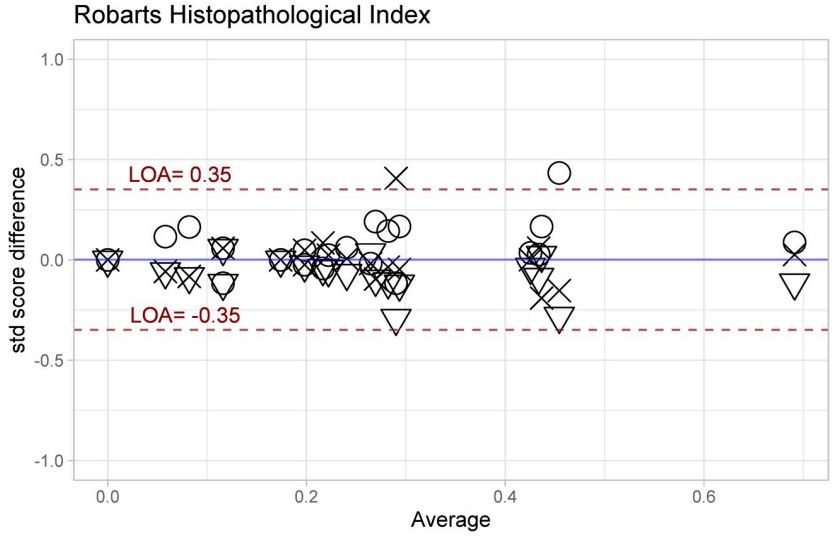

**Fig 2. Modified Bland-Altman plot.** The plot indicating the limits of agreements for the Final grade for each index. The plot shows less dispersion in the high and low average values, indicating higher agreement. The absolute scores were standardized then transformed by the square-root. This makes the LOA directly comparable.

The results show discrepancies in the severe inflammatory features due to low number of samples. This is not the case for the eosinophilic feature of GS and the structural architectural changes features. The number of eosinophils vary greatly between subjects depending on age and location in a healthy colon [37]. There is no recommendation for what an acceptable cut-off for eosinophils per segment of colon in UC remission is, and therefore, evaluating this feature coherently is challenging. Normal variation is also one of the challenges when evaluating structural architectural changes. Due to the inherent number of histological features included in this grade such as crypt branching, mucin depletion etc., an overview of it will leave too many features to subjective interpretations. Especially, since "grade 0.0 is indicated the absence of any abnormality.", which is almost never the case. It could be interpreted as disease specific abnormality, but the need for subjective interpretations on numerous features will challenge coherency between raters. GS is the only index to include such a feature.

Another subgrade that scored low in the inter-rater score is the Nancy Grade 1, chronic inflammatory cells. It is difficult to distinguish between moderate to severe amount of chronic inflammation from acute inflammatory features. There are seldom signs of severe chronic inflammation without concomitant presence of neutrophils, which defines the criteria of grade 1: "Grade 1 corresponds to the lack of mucosal neutrophils, a pivotal marker of disease activity, even though moderate or severe chronic inflammation can be present" [20]. Thus, making it a cause for variation and in many cases redundant.

Our intra-rater evaluation was as good or better than the inter-rater values, except for the eosinophile feature. Interestingly, there was a clearer difference between the raters than between modalities (white light microscopy or digitally scanned slides), suggested by the intra-rater results and the Kruskal-Wallis test (S1 Fig). This indicates that these methods can be used interchangeably for NI and RHI which does not evaluate eosinophils specifically. The high intra-rater score and the relatively low inter-rater score indicate that a central raters approach to IBD-pathology could be beneficial. Standardization of extraction, preparation and scanning is easier to achieve than extensive training of pathologists.

The modified Bland-Altman analysis identified the same as the Kruskal-Wallis analysis, that one pathologist rates higher than the two others, nevertheless the standard deviations are similar (S1 Table). The obvious explanation the IBD-related experience difference between LBR and the general pathologists. This indicating that experience gives a different understanding of the scores, which seems to effect accuracy but not necessarily precision as the IQR/SD are similar between raters. The LOA gives the amount the raters can be discordant with the mean estimated score. The results show better agreement for RHI than NI and GS. This could be a result of RHI being developed from GS by selecting the features of best agreement. If we evaluate the absolute LOA scores it shows that all the scores are rather insensitive to minor

**Table 3. Inter-rater agreement evaluated with Fleiss' kappa.**

|       | Strict | Relaxed |
|-------|--------|---------|
| **GS**  | 0.30 | 0.57 |
| **NI**  | 0.44 | 0.44 |
| **RHI** | 0.48 | 0.47 |

GS, Geboes Score; NI, Nancy Index; RHI, Robarts Histopathological Index.

differences, this can explain the drop in agreement when dichotomizing the indices from continuous variables to "Active" and "Remission". The agreement appears to be better in the low and high average scores for all three indices. An explanation could be that it is easier to rate the extremes of the distribution rather than the middle. This is unfortunate as the cut-off for remission is in the low-middle of the distribution.

In our data we defined two cut-off values for histologic remission in order to investigate whether there would be a difference between the two groups in relation to other clinical features of importance. There was no dependence between the clinical scores (MES and UCCS) and histologic category for any of the indices. This is important because a high degree of dependence between clinical scores and histology would render one of the factors redundant, as one factor could predict the other. By being independent they can complement each other Previous studies are conflicting in their report of this relationship between clinical scores and histologic scores [9, 12, 18, 21, 38].

## Strengths and limitations

Our study has several limitations, first and foremost is the different modalities used to evaluate the slides and the blinding to the biopsy location. The evaluation of eosinophils was challenged by two factors, one was the blinding of biopsy location to the pathologists and the different modalities of observation for the intra-rater analysis. Eosinophils significance in UC can be debated as the two recent indices does not it include it in a separate category and marked increase in eosinophils without other inflammatory cells suggest eosinophilic colitis and not UC. Despite the poor intra-rater value for eosinophils, the other categories had good intra-rater ICC, which suggests that error introduced by scanning the samples is small. Unfortunately, our biopsies are not orientated after collection so the pathologists could not reliably evaluate basal plasmacytosis, defined as plasma cells between the base of the crypts and *muscularis mucosae* [39]. Nevertheless, plasma cells fall under the category of chronic cell infiltrate, which is evaluated in all indices. One could argue that a scoring protocol is insufficient education to achieve accurate evaluation by a general pathologist.

Our strengths are the approximation of real-world setting where patients are under different treatment regimens and in different clinical settings, making any finding representative for the IBD remission population.

## Conclusion

Our study evaluated reliability of histology scores, in order to estimate their usefulness in clinical decisions. We found that there is a moderate to good agreement between raters when using three of the most common histological scoring indices, but with a LOA that could be improved. Unfortunately, when dichotomising the scores into active and remission the agreement falls to fair and moderate. Therefore, without more extensive training or the adoption of a central raters approach using the current histological indices for deciding remission should be done with caution.

## Supporting information

**S1 Fig. Difference between raters.** Significant difference between raters were tested with Wilcoxon rank sum test with Benjamini-Hochberg adjusted p-values.
(PDF)

**S2 Fig. Bar plot.** Difference in patients classified as remission or active according to relaxed or strict definition of remission.
(PDF)

**S3 Fig. Histologic score by biopsy location.** No difference was found between biopsy locations.
(PDF)

**S1 Table. Descriptics for raters.**
(DOCX)

**S2 Table. Kruskal-Wallis rank sum test on raters, by indices.**
(DOCX)

**S3 Table. Fisher exact test between the UCCS and MES scores and histologic category.** The table shows that histological category is independent for whether the sample was collected from a MES/UCCS 0 or 1 patient. This was true for both Strict and Relaxed category.
(DOCX)

**S4 Table. Wilcoxon rank sum test on the effect of medication on histological scores.**
(DOCX)

**S1 Appendix. Scoring aid provided to the pathologists.**
(DOCX)

**S1 Data.**
(CSV)

## Author Contributions

**Conceptualization:** Rasmus Goll.

**Formal analysis:** Christian Børde Arkteg, Rasmus Goll.

**Funding acquisition:** Jon Florholmen.

**Investigation:** Christian Børde Arkteg, Sveinung Wergeland Sørbye, Lene Buhl Riis, Stig Manfred Dalen.

**Methodology:** Sveinung Wergeland Sørbye, Rasmus Goll.

**Project administration:** Jon Florholmen.

**Supervision:** Jon Florholmen, Rasmus Goll.

**Writing – original draft:** Christian Børde Arkteg.

**Writing – review & editing:** Sveinung Wergeland Sørbye, Lene Buhl Riis, Stig Manfred Dalen, Rasmus Goll.

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
