## [Decision Letter · Decision Letter 0]

14 Dec 2020

PONE-D-20-32264

Real life evaluation of histologic scores for Ulcerative Colitis in remission

PLOS ONE

Dear Dr. Arkteg,

Thank you for submitting your manuscript to PLOS ONE. After careful consideration, we feel that it has merit but does not fully meet PLOS ONE’s publication criteria as it currently stands. Therefore, we invite you to submit a revised version of the manuscript that addresses the points raised during the review process.

We look forward to receiving your revised manuscript.

Kind regards,

Valérie Pittet, PhD

Academic Editor

PLOS ONE

2. In your Methods section, please provide additional information about the participant recruitment method and the demographic details of your participants. Please ensure you have provided sufficient details to replicate the analyses such as: the recruitment date range (month and year).

3. Please include the sequences of the primers used in your qPCR experiment in the supplementary data.

4. Thank you for including your ethics statement:  "The study and storage of biological material was approved of by the Regional Ethical Committee (REK Nord ID:2012/1349). "

Reviewers' comments:

Reviewer's Responses to Questions

**Comments to the Author**

1. Is the manuscript technically sound, and do the data support the conclusions?

Reviewer #1: Partly

Reviewer #2: Yes

Reviewer #3: Yes

2. Has the statistical analysis been performed appropriately and rigorously? 

Reviewer #1: I Don't Know

Reviewer #2: Yes

Reviewer #3: Yes

3. Have the authors made all data underlying the findings in their manuscript fully available?

Reviewer #1: Yes

Reviewer #2: Yes

Reviewer #3: No

4. Is the manuscript presented in an intelligible fashion and written in standard English?

Reviewer #1: Yes

Reviewer #2: Yes

Reviewer #3: Yes

5. Review Comments to the Author

Reviewer #1: The authors performed a timely study on the use of different histologic scoring systems to assess disease activity in patients with ulcerative colitis. They chose to study the three scoring systems that are most reported and recommended in literature. Biopsy samples from a total of 41 patients in clinical remission were scored independently by three general pathologists blinded to clinical and endoscopic findings. The authors show that the correlation between histologic disease activity and both clinical and endoscopic measures of disease activity was poor. This was also true for TNF levels measured by qPCR. Importantly, they show an inter-observer correlation (inter-class correlation) for the different scoring systems of 0.85, 0.73 and 0.70, respectively. This correlation was considerably lower when assessing histologic remission. The authors conclude that the relatively high inter-observer variation, compared to the available literature, may be due to differences in interpretation of the scoring systems between participating pathologists and/or the method used to calculate the variation.

The major strength of this study is that simulates a possible future clinical practice where not only gastrointestinal pathologists with an interest in IBD will be tasked to assess histologic remission status in biopsies from patients in clinical and endoscopic remission.

The ECCO has recently published guidelines on the use of histologic scoring systems. For clinical practice, they recommend to use the Nancy Index as the other scoring systems are deemed too complex in this setting. Please integrate this in the discussion.

It is common in articles describing inter-observer variation to inform readers on the experience of the pathologists. Please provide information regarding the length of experience and some insight into the number of IBD-related biopsies assessed annually.

From a biological point of view, the definitions of histologic remission are very different when comparing scoring systems. For instance, in the strict cut-off for histologic remission GS <2A does not correspond to NI =0 but rather to NI ≤1. Although the explanation by the authors is intuitive, the differences do not allow a valid comparison at the moment. Please adjust. Regarding the RHI <3, it is better to use an RHI ≤3 with subscores of 0 for lamina propria neutrophils and neutrophils in the epithelium.

The GS and the continuous GS are used interchangeably, which is confusing. Please use one system consistently.

The assessment of eosinophils is somewhat problematic. First, the fact that the pathologists were blinded to biopsy location may limit the assessment validity. Moreover, eosinophils were scored differently on scans compared to slides. Please address this in more detail in the discussion.

The authors state that intra-observer variation was assessed by one pathologist scoring the same biopsies on glass slides and using scans. Therefore, this variation cannot be quantified, as some (or indeed all) variation may be due to the difference between glass and scan. It does not seem appropriate to report the intra-observer variation in this context.

I miss a more thorough discussion on why the inter-observer variation was high compared to published data. I feel that sending four articles as a "scoring protocol" may serve as a common knowledge base, but is insufficient to introduce a new histologic parameter or scoring system in clinical practice. Moreover, the trend towards subspecialisation may result in dedicated gastrointestinal pathologists to be present in the majority of pathology institutes. I feel these important points warrant more discussion.

The authors performed “gene quantification” for TNF, as mentioned in the Materials in Methods section. I assume they measured mRNA levels. The integration of this data in the manuscript is limited. For instance, how many patients received anti-TNF therapy? Does this influence TNF mRNA levels? I do not feel this data adds value at the moment, consider removing it.

Please provide standard deviations and interquartile ranges where appropriate in Table 1.

Reviewer #2: In this manuscript, the authors evaluate the NHI, Geboes, and RHI among patients with clinical and endoscopic remission, in attempt to determine agreement between raters for each of the indices. Furthermore, they attempt to determine associations between clinical factors and histologic scores. Overall, the paper is trying to ask an important question about whether any of these indices have reliable inter-rater agreement, and furthermore, if the index rating can be further associated with clinical variables. However this paper is problematic for several reasons.

(1) The paper, overall, requires substantial revision to improve its readability, organization, and clarity. The introduction is wordy, and in many places several sentences are employed when one might have been used. Furthermore, the problem this paper plans to address (comparative utility of the three indices) is not well-set up, and the background and references need to be significantly strengthened, with a well-reasoned argument for why this problem is of importance. The materials/methods section suffers from a similar problem with some of the methods poorly defined. The histologic section, in particular, needs to be re-written with clarification of why specific cut-off values were chosen, further details on how pathologists rated slides (i.e. how many days separated each reading of the same slide?).

(2) The sample size here is small (41 patients) and inter-rater agreement has been previously assessed for these three indices using larger cohorts. Consequently, there is a problem of novelty here. What does this paper reveal that has not already been shown? The introduction should set this up with greater clarity so that the results section can address why this paper is novel, especially given the smaller sample size compared to prior efforts.

(3) The authors make the argument that assessment of histology is important for the ascertainment of clinical outcomes, however they do not evaluate clinical outcomes here. Instead, they examine cross-sectional clinical variables (MES 1 vs MES 0) and histologic grade and find no differences. Persistent histologic activity in MES 1 vs MES 0 has been well-established in larger cohorts; the failure to find a difference here is likely related to the smaller sample size. Given that, it remains unclear why these results should be of value in this draft. Furthermore, normalization of TNF levels was utilized as a surrogate for a predictor for long term remission; histologic grades were compared with high or low levels of TNF. This is an unusual covariate to examine, especially given the negative results here. I would raise questions as to why this covariate was specifically included, and would necessitate the authors include reasons as to why this - versus other potentially more useful covariates like calprotectin - was utilized.

Reviewer #3: This work is based on a cohort of UC patients in endoscopic remission and followed in one tertiary care center, whose aim was to assess histological activity based on the Geboes score, the Robarts and the Nancy indexes. I feel that some points need to be clarified.

- The authors observed lower concordance compared to the literature. The most convincing explanation is that the inclusion criteria used in this study differ from the literature, since only patients in endoscopic remission were included. It has been shown that histological markers of acute inflammation such as ulceration have a higher inter-reader agreement than markers of mild inflammation such as eosinophils. Please include this point in the discussion section.

- Since only patients in endoscopical remission were included, the results cannot be generalized to the overall UC population and the conclusion seems overstated (page 19 l. 272 'Therefore, using the current histological indices for deciding remission should be done with caution.') Please modify.

- It is not clear whether 41 patients (abstract) or 41 biopsies (page 13. l. 143) from X patients were assessed. Please clarify.

- In case of multiples biopsies in the same patient, how was assessed histological activity ? As the maximum histological activity from any biopsies ?

- Please include in the abstract that patients in clinical and endoscopical remission were included.

- It would have been interesting to have data on how long does it take to assess each score ? It might be important for pathologists, since the process could be time-consuming.

6. PLOS authors have the option to publish the peer review history of their article (what does this mean?). If published, this will include your full peer review and any attached files.

Reviewer #1: **Yes: **Aart Mookhoek

Reviewer #2: No

Reviewer #3: No

---

## [Author Response · Author response to Decision Letter 0]

28 Jan 2021

Reviewer 1

The ECCO has recently published guidelines on the use of histologic scoring systems. For clinical practice, they recommend to use the Nancy Index as the other scoring systems are deemed too complex in this setting. Please integrate this in the discussion.

- True, this will be incorporated, thank you for this suggestion. 

It is common in articles describing inter-observer variation to inform readers on the experience of the pathologists. Please provide information regarding the length of experience and some insight into the number of IBD-related biopsies assessed annually.

- Thank you for pointing this out, it will be addressed in the method chapter. 

From a biological point of view, the definitions of histologic remission are very different when comparing scoring systems. For instance, in the strict cut-off for histologic remission GS <2A does not correspond to NI =0 but rather to NI ≤1. Although the explanation by the authors is intuitive, the differences do not allow a valid comparison at the moment. Please adjust. Regarding the RHI <3, it is better to use an RHI ≤3 with subscores of 0 for lamina propria neutrophils and neutrophils in the epithelium.

- We agree that our strict for RHI were wrong and we have adjusted it as the reviewer suggested which is also in line with Pai et al. suggestion (Pai, 2018). The result is a slight improvement in RHI kappa Fleiss value. 

- Regarding the comparison between Geboes and Nancy we agree that the comparison is problematic, nevertheless we believe it has enough merit to withstand the problem. The main indicator for active inflammation is the presence of neutrophils. An abcense of neutrophils will equal to Geboes <2A.0 and Nancy = 0 because the presence of moderate to marked increase in chronic infiltrate is almost always followed by increase in neutrophils. In addition, ECCO’s position paper makes the same comparison (Margo,2020). Having the same cut of makes our results more comparative to this paper and some of Magro’s earlier papers (Journal of Crohn's and Colitis, 2020, 1–5 and Journal of Crohn's and Colitis, 2020, 169–175)

The GS and the continuous GS are used interchangeably, which is confusing. Please use one system consistently.

- Thanks for pointing this out, it is addressed in the final reviewed manuscript. All mentions of the Geboes score is now refered to the Geboes continous with the except of the ICC values. It is easier to understand the value of the ICC analysis if the indices are divided up in their subcategories and presented with their respective ICC value. In this way one can easily understand which sub-category has substantial variation. In order to facilitate the understanding when using them both we have provided a table in S2 appendix where the two indices are listed up against each other.

The assessment of eosinophils is somewhat problematic. First, the fact that the pathologists were blinded to biopsy location may limit the assessment validity. Moreover, eosinophils were scored differently on scans compared to slides. Please address this in more detail in the discussion.

- Yes this is a problem, and we have addressed it more thorough in our discussion.

The authors state that intra-observer variation was assessed by one pathologist scoring the same biopsies on glass slides and using scans. Therefore, this variation cannot be quantified, as some (or indeed all) variation may be due to the difference between glass and scan. It does not seem appropriate to report the intra-observer variation in this context.

- The reviewer makes a good point. In our approach it is difficult to distinguish what is the source of the variation observed. It is our argument that when the variation is small, even if two different modalities is used, it is unlikely that it would be greater if the same modality where used twice and therefore it is appropriate to present our results. Our intra-observer values are on par with what is reported previously (Mosli, 2017). 

I miss a more thorough discussion on why the inter-observer variation was high compared to published data. I feel that sending four articles as a "scoring protocol" may serve as a common knowledge base, but is insufficient to introduce a new histologic parameter or scoring system in clinical practice. Moreover, the trend towards subspecialisation may result in dedicated gastrointestinal pathologists to be present in the majority of pathology institutes. I feel these important points warrant more discussion.

- This is an interesting notion. The reason for this approach is to simulate the real world in hospitals outside the specialized central hospitals. There is, as the reviewer noted, a movement towards more subspecialisation, but this movement is slow and years will pass before there are GI-pathologist in the majority of hospitals in Scandinavia. Nevertheless, we see a clear difference between the general pathologist and the GI-specialized pathologists, which raises an interesting point of a centralized reader. The method of extraction, preparation and scanning could more easily be standardized. And, as we showed, with the exceptions of eosinophils, scanned biopsies do not introduce much variation. Therefore, we will modify our manuscript to introduce the suggestion of a centralized reader rather than extensive training of general pathologists. 

The authors performed “gene quantification” for TNF, as mentioned in the Materials in Methods section. I assume they measured mRNA levels. The integration of this data in the manuscript is limited. For instance, how many patients received anti-TNF therapy? Does this influence TNF mRNA levels? I do not feel this data adds value at the moment, consider removing it

• We agree that the TNF portion of the manuscript is a bit off topic, therefore we have removed any mention of it from the manuscript. 

Reviewer 2

The paper, overall, requires substantial revision to improve its readability, organization, and clarity. The introduction is wordy, and in many places several sentences are employed when one might have been used. Furthermore, the problem this paper plans to address (comparative utility of the three indices) is not well-set up, and the background and references need to be significantly strengthened, with a well-reasoned argument for why this problem is of importance. The materials/methods section suffers from a similar problem with some of the methods poorly defined. The histologic section, in particular, needs to be re-written with clarification of why specific cut-off values were chosen, further details on how pathologists rated slides (i.e. how many days separated each reading of the same slide?).

- Thank you for addressing this, we will work on improving the readability of the paper. We have revised the introduction and the method chapter to make it clearer, more readable and to the point. Thank you for helping us improve the overall quality of our paper. 

The sample size here is small (41 patients) and inter-rater agreement has been previously assessed for these three indices using larger cohorts. Consequently, there is a problem of novelty here. What does this paper reveal that has not already been shown? The introduction should set this up with greater clarity so that the results section can address why this paper is novel, especially given the smaller sample size compared to prior efforts.

- This is a valid argument. We believe that our focus on the remission patient and that the evaluation is performed by general pathologist as well as GI-specialized pathologists makes our paper different from these large cohorts. They tend focus on UC as a whole and not remission patients. In addition, the histology is evaluated by only expert or extensively IBD-trained pathologists. 

The authors make the argument that assessment of histology is important for the ascertainment of clinical outcomes, however they do not evaluate clinical outcomes here. Instead, they examine cross-sectional clinical variables (MES 1 vs MES 0) and histologic grade and find no differences. Persistent histologic activity in MES 1 vs MES 0 has been well-established in larger cohorts; the failure to find a difference here is likely related to the smaller sample size. Given that, it remains unclear why these results should be of value in this draft. Furthermore, normalization of TNF levels was utilized as a surrogate for a predictor for long term remission; histologic grades were compared with high or low levels of TNF. This is an unusual covariate to examine, especially given the negative results here. I would raise questions as to why this covariate was specifically included, and would necessitate the authors include reasons as to why this - versus other potentially more useful covariates like calprotectin - was utilized.

- The reviewer makes a very good argument and the issues surrounding the TNF data has been noted by another reviewer therefore we have decided to remove it from the manuscript. 

We mentioned in the introduction that histology needs to fulfil three criteria in order to be useful, it must be able to identify signs of inflammation, it must be reliable, and it must have an effect on outcome. We explicitly state that we will focus on the two first criteria. While larger studies than ours exist, with different results, we believe it does not negate our results as our focus is different. In addition, the association between histology score and MES is of importance to answer our first statement. If histology provides additional information it would not show any difference between MES 1 and 0 in a Fishers exact test for the following reason: If two factors separate a population in the exact same partitions(ie all MES 0 patients are in the histologic remission group) , then one of the factors are a surplus. When they are independent (ie not significant Fishers exact test), they can complement each other. Therefore, there is value in investigating whether the MES 1 and MES 0 population differ in regard to histology category. This point was a bit confusing in the original text. We have changed some of the statistic test to make it clearer in the revised text.

Reviewer 3

The authors observed lower concordance compared to the literature. The most convincing explanation is that the inclusion criteria used in this study differ from the literature, since only patients in endoscopic remission were included. It has been shown that histological markers of acute inflammation such as ulceration have a higher inter-reader agreement than markers of mild inflammation such as eosinophils. Please include this point in the discussion section.

- Thank you for your input, your notion will be incorporated in the final manuscript. 

Since only patients in endoscopic remission were included, the results cannot be generalized to the overall UC population and the conclusion seems overstated (page 19 l. 272 'Therefore, using the current histological indices for deciding remission should be done with caution.') Please modify.

- This is true we will make adjustment to the manuscript. Thank you for making us aware of this problem.

- It is not clear whether 41 patients (abstract) or 41 biopsies (page 13. l. 143) from X patients were assessed. Please clarify.

- This will be clarified in the final draft, thank you for bringing this to our attention. 

- In case of multiples biopsies in the same patient, how was assessed histological activity? As the maximum histological activity from any biopsies?

- This is explained under the subheading “Histology” and the lines 89-90. 

- Please include in the abstract that patients in clinical and endoscopic remission were included.

- Thank you for aiding us in improving the paper, we will improve the abstract. 

- It would have been interesting to have data on how long does it take to assess each score? It might be important for pathologists, since the process could be time-consuming.

- This is a very interesting point, but unfortunately, we do not have the time used for evaluating each slide. All the pathologist agreed that they used the longest time on Geboes and the shortest time on Nancy, but we cannot confirm this with data.

---

## [Decision Letter · Decision Letter 1]

23 Feb 2021

Real life evaluation of histologic scores for Ulcerative Colitis in remission

PONE-D-20-32264R1

Dear Dr. Arkteg,

We’re pleased to inform you that your manuscript has been judged scientifically suitable for publication and will be formally accepted for publication once it meets all outstanding technical requirements.

Kind regards,

Valérie Pittet, PhD

Academic Editor

PLOS ONE

Additional Editor Comments (optional):

Reviewers' comments:

Reviewer's Responses to Questions

**Comments to the Author**

1. If the authors have adequately addressed your comments raised in a previous round of review and you feel that this manuscript is now acceptable for publication, you may indicate that here to bypass the “Comments to the Author” section, enter your conflict of interest statement in the “Confidential to Editor” section, and submit your "Accept" recommendation.

Reviewer #1: (No Response)

Reviewer #3: All comments have been addressed

2. Is the manuscript technically sound, and do the data support the conclusions?

Reviewer #1: Yes

Reviewer #3: Yes

3. Has the statistical analysis been performed appropriately and rigorously? 

Reviewer #1: Yes

Reviewer #3: Yes

4. Have the authors made all data underlying the findings in their manuscript fully available?

Reviewer #1: Yes

Reviewer #3: Yes

5. Is the manuscript presented in an intelligible fashion and written in standard English?

Reviewer #1: Yes

Reviewer #3: Yes

6. Review Comments to the Author

Reviewer #1: Thank you for addressing my comments concerning the manuscript. I feel that the changes made based on the comments from the different reviewers have improved the manuscript.

I continue to disagree with the argumentation regarding the use of a Nancy Index score of 0 as the definition of histologic remission. In my experience, a Nancy Index of 1 is not so rare. However, I agree with the authors that this definition has gained some traction in recent literature.

Reviewer #3: Thank you for providing the extensive revisions requested by the editor and both reviewers. I feel that my queries have been adequately addressed.

7. PLOS authors have the option to publish the peer review history of their article (what does this mean?). If published, this will include your full peer review and any attached files.

Reviewer #1: **Yes: **Aart Mookhoek

Reviewer #3: No

---

## [Editor Report · Acceptance letter]

26 Feb 2021

PONE-D-20-32264R1 

Real-life evaluation of histologic scores for Ulcerative Colitis in remission 

Dear Dr. Arkteg:

I'm pleased to inform you that your manuscript has been deemed suitable for publication in PLOS ONE. Congratulations! Your manuscript is now with our production department. 

Kind regards, 

on behalf of

PD Dr. Valérie Pittet 

Academic Editor

PLOS ONE